# (GIGA)byte

TECHNICAL RELEASE

# CannSeek? Yes we Can!
# An open-source single nucleotide polymorphism database and analysis portal for *Cannabis sativa*

Locedie Mansueto[1], Kenneth L. McNally[2], Tobias Kretzschmar[1] and Ramil Mauleon[1,2,*]

1 Southern Cross University, Military Road, Lismore New South Wales 2480, Australia
2 International Rice Research Institute, Pili Drive, Los Baños Laguna 4031, Philippines

## ABSTRACT

A growing interest in *Cannabis sativa* uses for food, fiber, and medicine, and recent changes in regulations have spurred numerous genomic studies of this once-prohibited plant. Cannabis research uses Next Generation Sequencing technologies for genomics and transcriptomics. While other crops have genome portals enabling access and analysis of numerous genotyping data from diverse accessions, leading to the discovery of alleles for important traits, this is absent for cannabis. The CannSeek web portal aims to address this gap. Single nucleotide polymorphism datasets were generated by identifying genome variants from public resequencing data and genome assemblies. Results and accompanying trait data are hosted in the CannSeek web application, built using the Rice SNP-Seek infrastructure with improvements to allow multiple reference genomes and provide a web-service Application Programming Interface. The tools built into the portal allow phylogenetic analyses, varietal grouping and identifications, and favorable haplotype discovery for cannabis accessions using public sequencing data.

**Availability and implementation:** The CannSeek portal is available at https://icgrc.info/cannseek, https://icgrc.info/genotype_viewer.

**Subjects** Software and Workflows, Bioinformatics, Agriculture

**Submitted:** 18 June 2024

* Corresponding author. E-mail: ramil.mauleon@scu.edu.au

Preprint submitted at https://doi.org/10.25918/preprint.367

## STATEMENT OF NEED

### Background on *Cannabis sativa*

*Cannabis sativa* (cannabis, NCBI:txid3483) is an ancient, versatile and highly plastic crop that has been used for food, fiber and medicine for millennia. The first evidence of cultivation for use as fiber for ropes, textiles, and paper is from China, dating back to 4000 B.C. The use as medicine (for rheumatic pain, constipation, female reproductive disorders and malaria) by the ancient Chinese was reported in the Pen-ts'ao Ching, the first pharmacopeia from 2700 B.C. It was also used as an anesthetic during surgery in 110–207 AD [1, 2]. Early nomads from Central Asia helped spread cannabis to Europe around 4000 years ago, and Arab traders introduced it to Africa around 2000 years ago [3]. While it was spread globally and used extensively throughout the 18th and 19th centuries, playing a crucial role in the colonial expansion of European nations, the use of *C. sativa* became largely prohibited worldwide throughout the second half of the 20th century, classified as a narcotic drug, and enforced through the United Nations Single Convention

Treaty on Narcotic Drugs [4]. Over the last decade, however, there has been a global resurgence in cannabis use, spearheaded by the same nation that enforced its widespread prohibition.

There are clear legislation and market-driven distinctions between low-tetrahydrocannabinol (THC), industrial hemp (for food and fibers), and high-THC medicinal cannabis (for medicinal and recreational purposes). Industrial hemp is now a global multibillion-dollar industry with a diverse product range competing with oil, protein, biomass and fiber crops in diverse markets [5]. Equally, medicinal cannabis has quickly grown into a multibillion-dollar industry with a multitude of medicinal applications and a relaxation of regulation on its recreational use. However, due to its widespread prohibition in the 20th and early 21st century, *C. sativa* has missed out on many technological advances that have greatly benefited the improvement of traditional crops [6, 7].

## Survey of *Cannabis* genomics datasets

The changes in regulation around *C. sativa,* combined with public interest and developments in instrumentation and informatics, have spurred unprecedented expansion of *C. sativa* genomic research and development over the last decade [8]. Cannabis genome assemblies have been made available with chromosome level builds, including Purple Kush (PK) (the first published genome for medicinal use), Finola (FN) (hemp), CS10 (medicinal), Cannabio-2 (medicinal) and JL (wild) [9–12]. The CS10 hybrid assembly (RefSeq GCF_900626175.2) has been the gold-standard reference available at NCBI.

A wealth of resequencing and high-density genotyping data has been recently generated, mostly to perform diversity and phylogenetic analyses (340 samples from Lynch *et al.* [13], 40 cultivars from McKernan *et al.* [14], 100 accessions from Ren *et al.* [15], and the German Genebank (IPK) collection from Woods *et al.* [16]. Transcriptome data were also available from RNA-Seq experiments used for single nucleotide polymorphism (SNP) discovery (Booth *et al.* [17], Zager *et al.* [18], Livingston *et al.* [19], and Braich *et al.* [20]). Commercial entities also contribute to genotyping datasets, albeit using low coverage resequencing that targets cannabinoid synthases (Phylos dataset NCBI PRJNA347566, PRJNA510566; European Variation Archive PRJEB49958 vs. Cannatonic reference). Kannapedia also has an explorable phylogenetic tree [21], with data derived from targeted sequencing, published whole genome sequencing, and their genotyping chip. To demonstrate the utility of commercial databases, Aardema *et al.* [22] used datasets downloaded from these resources and, using genome-wide association studies (GWAS), found they may have sufficiently high-quality information for most interesting traits like THC and cannabidiol (CBD) content. They also found genotype correlation for other chemical and agronomic traits, but less reliable on subjective effects from usage. Using nine commercial collections, Halpin-McCormick *et al.* [23] found inconsistencies in the population structures based on use type, but observed structure based on geo-reference. The authors also observed broader genetic diversity in modern cultivars compared to landraces.

## Motivation for *Cannabis* genotype database

There is clear value in consolidating and integrating existing public data in order to increase the power of these discovery analyses. An identified gap is the coordinated approach to unifying/consolidating these isolated genotyping datasets in a data portal with a single query point and a user-friendly graphical interface. Sites are hosting public

Cannabis datasets, like CannabisGDB [24] for gene loci prediction, proteins and metabolites. CoGE [25] is a general database for comparative genomics for genomes mostly taken from NCBI. However, they also host Cannabis genomes and resequencing data not available in NCBI (e.g., JL DASH, First Light and JL Mother). None of these sites hosts public genotype datasets for cannabis, especially in a readily usable format. Furthermore, genome variant mining tools are notably absent in these resources.

Web applications to host large genotype datasets like Gigwa, MaizeGDB SNPversity, and KnowPulse Tripal Genomic Variation have been used in crops [26–28]. After the release of sequences from the 3000 rice genomes project [29], SNP-Seek [30–32] was developed to provide the rice research community with a comprehensive tool to mine this resource for allelic variation relative to reference genomes. Given the utility and popularity of SNP-Seek for rice, we chose the SNP-Seek software infrastructure to host a genotyping data portal for cannabis.

## IMPLEMENTATION
### Variant data generation

We utilized publicly available *Cannabis* sequences and community-standard variant discovery methods to generate the genotyping dataset for the portal.

#### Resequencing data sources

Without traceable gene bank passport information, we had to rely on the limited sample descriptions available from publications and NCBI BioSample (RRID:SCR_004854) entries. We restricted this study to the available information on plant use or type (e.g., hemp, medicinal, or CBD), which is provided by most of the studies. Some studies also include data on terpenoids and cannabinoids.

A summary of the samples we used is presented in Table 1. Most of the samples are from multi-varieties next-generation sequencing (NGS) whole-genome resequencing projects to study cannabis diversity. Some are from genome resequencing projects, others from targeted resequencing (e.g., Phylos) and RNA-Seq. Three genotype matrices were generated for each reference: whole genome sequencing-7 sources (WGS7DS), trichome RNA-Seq 26 samples (26TRICH), and PHYLOS amplicon. For WGS7DS, the number of samples for each cultivar type based on published data is summarized in Table 2.

#### Reference genomes used

Three genomes, cs10, PK and FN [9, 10, 36], were used as references for variant calling. They represent the Cannabis diversity for CBD, drug, and hemp types. Cs10 was annotated with genes from NCBI RefSeq (RRID:SCR_003496), while PK and FN have publicly available RNA-Seq data for gene prediction.

#### Variant-calling workflow

GATK (RRID:SCR_001876) [37] and NVIDIA Parabricks [38] germline pipelines were used for variant discovery, and were benchmarked using the cs10 assembly. With comparable results but significantly faster run times, as determined by our benchmarking studies [39], Parabricks was used for all genome assemblies (cs10, PK, and FN). Resequencing datasets were either genomic or RNA-Seq. NGS FASTQ sequences were downloaded from NCBI Sequence Read Archive (SRA) (RRID:SCR_004891) and Kannapedia [21]. Trimmomatic



**Table 1.** Publicly available NGS datasets used for cannabis SNP discovery.

| Dataset | Reference | Cultivars |
|---|---|---|
| WGS7DS – 7 whole genome NGS datasets | [13] 10.1080/07352689.2016.1265363 PRJNA310948 | (55) Afghan_Kush_1, Afghan_Kush_2, Afghan_Kush_3, Afghan_Kush_4, Afghan_Kush_5, Afghan_Kush_6, Alaskan_Thunderfuck, Auto_AK47, B-5, Blue_Dream_3, Blueberry_DJ, Cannatonic, Carmagnola_1, Carmagnola_2, Carmagnola_3, Carmagnola_4, Carmagnola_5, Carmagnola_6, Chem91, Chinese_hemp, Chocolope_1, Dagestani_hemp, Durban_Poison_1, Durban_Poison_2, EuroOil_2, Feral_Kansas, Feral_Nebraska_1, Feral_Nebraska_3, G13, Girl_Scout_Cookies_1, Golden_Goat_2, Grape_Ape_1, Harlequin, Hawaiian, Hindu_Kush, Jack_Herer_1, Kompolti_1, Kompolti_2, Kunduz, Lebanese, Liberty_Haze, Low_Ryder, Maui_Waui, OG_Kush, Original_Sour_Diesel, Pre-98_Bubba_Kush, R4, Rocky_Mountain_Bluberry, Sievers_Infinity, Skunk_#1, Somali_Taxi_Cab, Super_Lemon_Haze, Tangerine_Haze, Tora_Bora, White_Widow_1 |
| | [14] 10.1101/2020.01.03.894428 PRJNA575581 | (40) 80 E-1, 80 E-2, 80 E-3, Arcata Trainwreck, Black 84, Black Beauty, BlueBerry Cheesecake X JL Male, C3/USO-1_F1_15_CSU, Carmagnola_3, Carmaleonte, Chem 91, Citrix, CS_1_2016_CSU, Domnesia, Eletta Campana, Fedora17_6_1_CSU, Grape Stomper, Harlox, Headcheese, Herijuana, IdaliaFT_1_CSU, Jamaican Lion ˆ4 #1, Jamaican Lion ˆ4 #2, Jamaican Lion ˆ4 #3, Jamaican Lion ˆ4 #4, Jamaican Lion ˆ4 #5, Jamaican Lion ˆ4 #6, Jamaican Lionˆ3 Father, Jamaican Lionˆ3 Mother, Jamaican Lionˆ3 Mother PCR, Master Kush, Merino_S_1_CSU, Mothers Milk #5, Red Eye OG, Saint Jack, Sour Diesel, Sour Tsunami, Sour Tsunami x Cataract Kush, Tahoe OG, Tiborszallasi |
| | Kannapedia https://medicinalgenomics.com/kannapedia-fastq | (58) AK47, AfghanKush, Afgooey, AlaskanIce, ArcataTrainWreck, ArjanUltraHaze2, AustralianBastard, BlueBerryCheeseCake18, BlueBerryCheeseCakeBC2Fem, BlueBerryEssense, BlueDreamSCC, Breakthrough, C4XCanatsuSCC, CBDMangoHaze, CanaTsuSCC, CaseyJones, CheeseGHS, ChemDawg91, ChemDawg, ChemDog18cycles, ChemDog, ChemdogXCherryPieSCC, Cinex, DakiniKushMale, DeepPurpleHaze, DiamondGirl, EastCoastSourDiesel, FireOG, G4XSFMSCC, GirlScoutCookie, GrapeStomper, GreenCrackSCC, Haleys, JackHerer, JambaCity, KushITSCC, LuckyCharms, MicheNepalMale, MoonshineHaze2, OGKushSCC, OGKushTest1, OGXPKSCC, PKX808OGSCC, PureKush, RedDevil, Ringo, RioNegraMale, RollexKush, SSHXWWSCC, SecretOG, SensiStarXSFMSCC, SnoopDream, SourTsunami, SuperLemonHaze, TrainwreckSCC, WIFIGTUBE, WIFI, WZ, Watermelonhazemale, WhiteWidow, WonderWoman, YeddiMale |
| | [15] 10.1126/sciadv.abg2286 PRJNA734114 | (82) Uniko B HUO, Fibranova IFA, Kompolti HKI, Beniko PBO, Carmagnola 2 ICA2, Tiborszallasi HTI, Big Bud BBD, Big Skunk BSK, Delta-llosa SDA, Swaziland SWD, Ruderalis Indica RIA, Top 44 TOP, Northern Light NLT, Alpine Rocket ART, Haze HAE, Mexican Sativa MSA, Hawaii Maui Waui HMW, PP9, Hindu Kush HKH, Juso14 UJO, IUP1, IUP2, IUP3, B52, IUL1, IUL2, IUL3, IBR1, IBR2, IBR3, PID1, PID2, PCL1, PCL2, Bialobrzeskie PBE, VIR 469-1 KAK1, VIR 469-3 KAK3, VIR 469-2 KAK2, VIR 483-1 UTT1, VIR 483-2 UTT2, VIR 483-3 UTT3, R2in135-1 NER1, R1in136-1 ERM1, R3in134-1 NEB1, VIR 37, Novgorod-Seversky, cv UNS, Ferimon 12 FFN, VIR 201 UKE, VIR 369 BUA, VIR 493, Glukhovskaja 10 Zheltostebel'naja UGA, VIR 507, Krasnodarsky 10 FB RKY, IBE, R1in136-2 ERM2, R1in136-3 ERM3, R2in135-2 NER2, R2in135-3 NER3, R1in136-4 ERM4, Fedora 17 FFA, R3in134-2 NEB2, R2in135-4 NER4, R3in134-3 NEB3, VIR 223, Bernburgskaya Odnodomnaya, bm GBA, R3in134-4 NEB4, Wild Thailand THD, missing PEU, Colombian 8 COA, VIR 449, Szegedi 9 HIS, XHC1, Santhica 27 FSA, XHC2, XGL1, XGL2, XBL1, XBL2, XUM1, IMA, XUM2, SCN, QHI, Carmagnola 1 ICA1, YNN, GXI, Chamaeleon NCN |
| | From various NGS sequencing projects | [34] 10.1038/s41598-020-75271-7 PRJNA669610 (2) CBDA pool, THCVA pool [33] 10.1093/genetics/iyab099 PRJNA723060 (3) Carmagnola, USO31, Carmagnola x USO31 |
| | [16] PRJNA866500 10.1093/g3journal/jkac209 | (135, w/ replicates) Bialobrzesk, Carmagnola, Carmealon, Dac, Dia, Eletta Campa, Fedora_17, Felina, Ferimo, Futura, IPK_100, IPK_16, IPK_17, IPK_18, IPK_19, IPK_20, IPK_21, IPK_22, IPK_23, IPK_24, IPK_26, IPK_27, IPK_28, IPK_29, IPK_30, IPK_31, IPK_32, IPK_33, IPK_34, IPK_35, IPK_36, IPK_37, IPK_38, IPK_39, IPK_40, IPK_41, IPK_42, IPK_43, IPK_44, IPK_45, IPK_46, IPK_48, IPK_49, IPK_50, IPK_51, IPK_52, IPK_53, IPK_54, IPK_55, IPK_56, IPK_57, IPK_58, IPK_59, IPK_60, IPK_61, IPK_63, IPK_64, IPK_65, IPK_68, IPK_69, IPK_70, Jiang, Lovr, Meng, Monoi, Santhica, Tiborszalla, Tis, Tyg, US_feral, USO_31 |
| | From genome assembly projects | CBDRx, Purple Kush, Finola, Jamaican Lion-DASH, and JL |
| PHYLOS | BioProjects PRJNA347566 PRJNA510566 | 2,223 cultivars |
| 21TRICH 21 RNA-Seq sequences collected from trichomes | [18] PRJNA498707 | Sour Diesel, Canna Tsu, Black Lime, Valley Fire, White Cookies, Mama Thai, Terple, Cherry Chem, Blackberry Kush |
| | [17] PRJNA599437 | Afghan Kush, Blue Cheese, CBD Skunk Haze, Chocolope, Lemon Skunk |
| | [19] PRJNA483805 | Finola bulbous, sesille, stalked |
| | [20] PRJNA560453 SRR10600904, SRR10600906, SRR10600907, SRR10600908, SRR10600912, SRR10600913, SRR10600916, SRR10600918, SRR10600920, SRR10600922, SRR10600923, SRR10600925 | Cannbio-2 trichome, four development stages, three replicates |
| 26TRICH – 21TRICH update | [35] PRJNA706039 | chemdawg, headband, ghost_ogxbk, tahoe_ogxbk, westside |

**Table 2.** Summary of cultivar types for the Whole-Genome dataset WGS7DS.

| WGS7DS cultivar types | Sub-type | No. of sample |
|---|---|---|
| Hemp | Hemp-type | 132 |
| | Type III | 9 |
| Drug | Drug-type | 17 |
| | NLDT | 26 |
| | BLDT | 12 |
| | Drug-type feral | 17 |
| | Type I | 29 |
| Basal cannabis | | 14 |
| Type II | | 17 |

(RRID:SCR_011848) [40] was used to trim adapters before running the variant calling pipelines. Following the GATK Germline Pipeline for the genomic sequences [41], BWA-MEM2 (RRID:SCR_022192) [42], GATK MarkDuplicate was run to generate the BAM alignment file for each sample. BioSamples with multiple sequence samples were merged using GATK MergeSAM before using GATK HaplotypeCaller to generate GVCF files. GenomicsDB was used for merging all sample GVCF, then GenotypeGVCF to generate the VCF file with parameters *–– heterozygosity 0.013 –– indel-heterozygosity 0.0013*, based on result from a previously analyzed subset.

Variant calling for a large number of accessions is notoriously slow using GATK HaplotypeCaller-based pipelines. To speed this up, we explored accelerated methods for variant calling. Parallel runs were performed to compare the speed, resource usage, and accuracy between GATK and Parabricks. The optimized workflow in Figure 1 incorporating Parabricks includes the following steps: (1) fq2bam: each NGS sequence read was aligned to the reference; (2) haplotypecaller: the variants were discovered by distributed runs by sample and chromosome; (3) GenomicsDBImport: all samples were combined into a GenomicsDB database by chromosome; (4) GenotypeGVCF: joint genotyping was performed to get dataset-wide statistics and variants; finally, (5) GatherVCFs: all chromosome variants were concatenated at the end for the resulting whole genome. Each step was performed by a software module of either GATK for the CPU version, or Parabricks for the GPU version. With a considerable improvement in speed and minimal difference in results, we used Parabricks over GATK on the discovery for all the references.

For the RNA-Seq pipeline based on the GATK pipeline presented in Figure 1 (top right), STAR (RRID:SCR_004463) [43], GATK MarkDuplicate, SplitNCigar and MergeSAM were used to generate the BAM files, followed by variant calling as described above. Similarly, Parabricks rna_fq2bam was run for the benchmarking study. The RNA-Seq variant calling is similar to that for genomic except for an intermediate correction step on the alignment results using SplitNCigarReads to account for exons.

Running Parabricks is straightforward using pbrun fq2bam and pbrun haplotypecaller. Parabricks was used for its speed advantage (15–20× faster) and lower usage (5× less) of computing service units. However, the merging of sample GVCF files was done using legacy GenomicsDB as Parabricks does not generate muti-sample GVCF with a large sample size. GATK GentotypeGVCF was then used for joint genotyping using the generated GenomicsDB databases. The implementation of this workflow is described in Protocol 1 in [44].

The SNPs were then separated from the indels using the BCFtools (RRID:SCR_005227) [45] norm and filter functions with this pipeline:



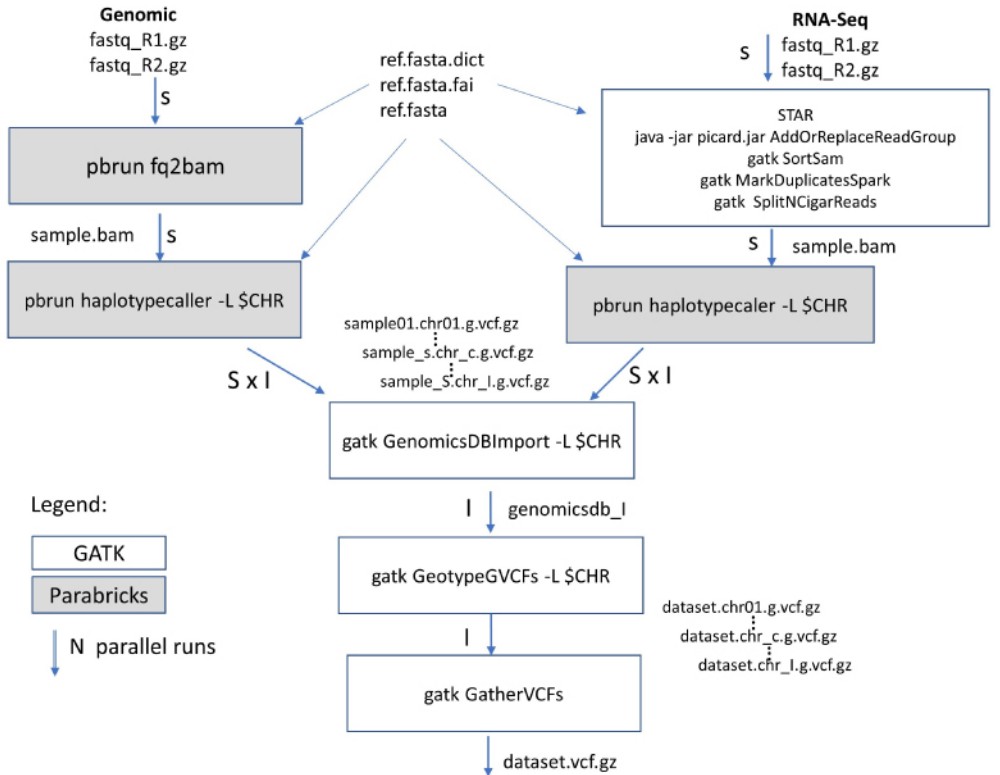

**Figure 1.** **Genomic and RNA-Seq Variant calling pipeline using GATK and Parabricks.**
Parabricks was used in the whole genome sequences, while GATK was used in the RNA-Seq variant calling of samples. GATK GenomicsDB was used to merge and genotype all samples jointly. The pipeline is optimized for the shortest wall time while maintaining sensitivity using high-performance computing environments with graphic processing units. Tasks are split and distributed whenever possible by the S samples and I genomic intervals. Each chromosome is considered as one interval, while all unassembled contigs are combined as one interval.

```
bcftools norm --atom-overlaps . -c w -a --fasta-ref reference.fasta input.vcf.gz
| bcftools filter -i 'TYPE="snp"'| bcftools norm -m +any -Oz -o
output.snpsonly.vcf.gz
```

Finally, the SNP attributes were recomputed using bcftools +fill-tags to generate the datasets for the analyses below and loaded into the CannSeek database.

## SNP data storage and retrieval

Detailed implementation of this section is described in Protocol 2 in GigaDB [44]. SNP data are hosted in CannSeek (RRID:SCR_025579), an SNP-Seek instance for fast query and visualization. The generated VCF file was transformed into an SNP matrix using a bcftools query, then to HDF5 using customized software that utilizes the HDF5 library. The web application features the Genotype Query page to retrieve a subset of the genotype matrix constrained by dataset and chromosome region. The SNP query result can be filtered further for synonymous or non-synonymous, sample name, dataset, or allele values in positions. The matrix is displayed on the application and can be downloaded in various formats (e.g., csv, tab-delimited, Flapjack, or PLINK).

As an external application that is part of the ICGRC Tripal server [46], CannSeek Apache Tomcat and its Postgres database run inside separate Docker/podman containers.

## CannSeek database

The SNPs were loaded into the CannSeek database, derived from the Rice SNP-Seek Database [32], and made available in the Genotype Viewer [47] as part of the ICGRC portal. This allows easy and fast access to variant datasets for the cannabis research community. Figure 2 shows the CannSeek query interface. The datasets are first selected by the 'Reference' used, 'Cultivar set', and 'SNP set', which is basically the SNP matrix of samples (cultivar) by SNPs called against the selected reference assembly. The 'Cultivar set' options depend on the selected 'Reference', and the 'SNP set' options depend on the selected 'Cultivar set'. Currently available are the [cs10, pkv5, fnv2] references, and the [6_wgs,7_wgs,phylos,trichomes_rnaseq,trichomes26_rnaseq] cultivar sets for all references. Genotype data publicly shared by publications were also loaded similarly to Woods *et al.* [16] for cs10. Details of the different datasets are listed on the CannSeek page. Multiple cultivar sets can be selected where the resulting genotype matrix is a concatenation of sample rows, and the SNP columns are the union or intersection depending on the 'Combine by' option. This is convenient for verifying the allele calls made on the same sample from different matrices, like the shared samples in the WGS6DS, WGS7DS, and Woods *et al.* sets. The SNPs to display could be "All", "All with highlighted nonsynonymous", or "Nonsynonymous only", where synonymy is based on the gene models by RefSeq for cs10, and the predictions by FINDER for pkv5 and fnv2. Once the dataset is selected, the region is constrained by the 'Chromosome', 'Start', and 'End' base positions. The positions can also be filtered by choosing from the 'Gene locus' options. The query result is the SNP matrix that satisfies the query options. The allele or genotype frequencies of SNP are plotted at the bottom of the matrix. The matrix displayed is downloadable in csv, tab, plink, or Flapjack formats. The haplotype viewer in Figure 3 represents the result matrix with the samples reordered after clustering. The color-coding of alleles as reference homozygous (gray), alternate homozygous (red), and heterozygous (gray) gives an instant visual overview of the genomic diversity of the queried region, while the clustering of alleles indicates the different haplotype blocks. The clustering calculation details and the resulting cluster alleles are also provided.

## Data download and API

The ICGRC Omics API is designed to programmatically access cannabis-omics datasets by web services (see accompanying paper ICGRC [48]) and is documented in the icgrc.info site menu (Infome:Omics API Documentation). This includes the variant datasets generated in this project. Users can customize their SNPs filter criteria using the Omics API (/user/variant/{keyword_region}), guided by the information provided in the frequency distribution plots of the SNP properties shown in Figures 4 and 5.

For genome-wide studies, however, it is recommended to download the PLINK-formatted data from the ICGRC Download page [49] and use it as shown in the accompanying Jupyter notebooks and Python modules in the ICGRC Omics API documentation [50] and demonstrations [51, 52].

## SNP analyses and visualizations
### *Haplotype blocks*

Haplotype blocks were detected by clustering the samples based on the similarity of SNPs. SNPs were clustered using the hamming distance, a tree constructed using R statistical



# CannSeek (Cannabis SNPs Genotype Viewer)

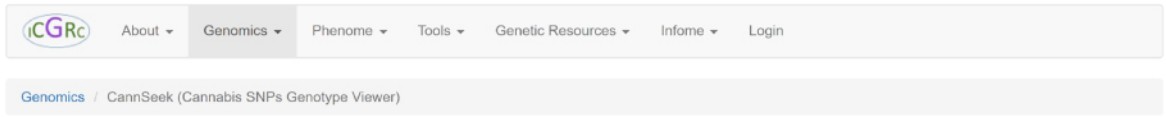

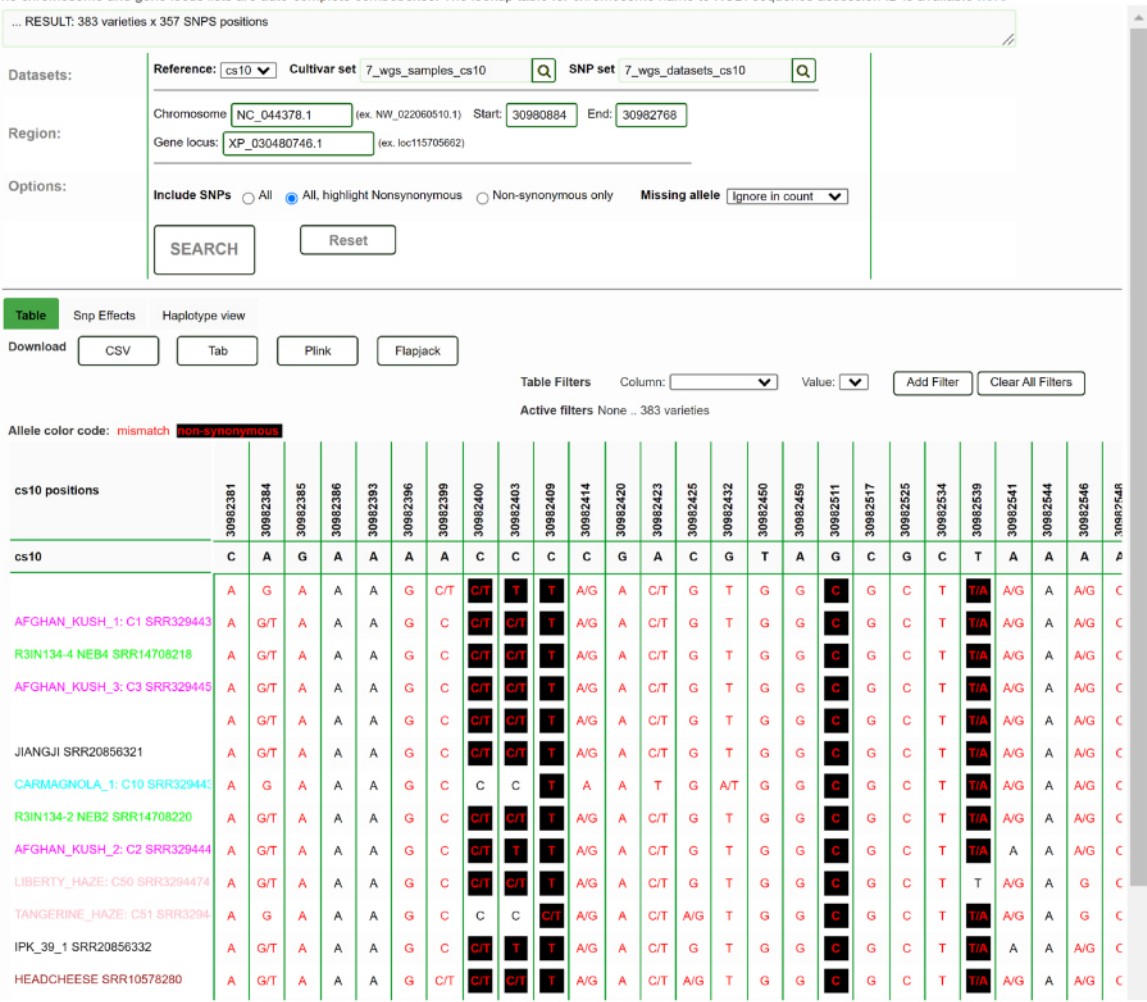

**Figure 2. CannSeek SNPs query interface.**
The CannSeek genotype viewer [47] is an interactive web application to query a subset of the large genotype matrices. The user specifies the dataset, genomic region, and display options. The alleles are color-coded to indicate reference match, and the synonymous or non-synonymous mismatch. Further filtering by column value can be performed on the result. The matrix can also be downloaded into csv, tsv, flapjack, or plink formats.

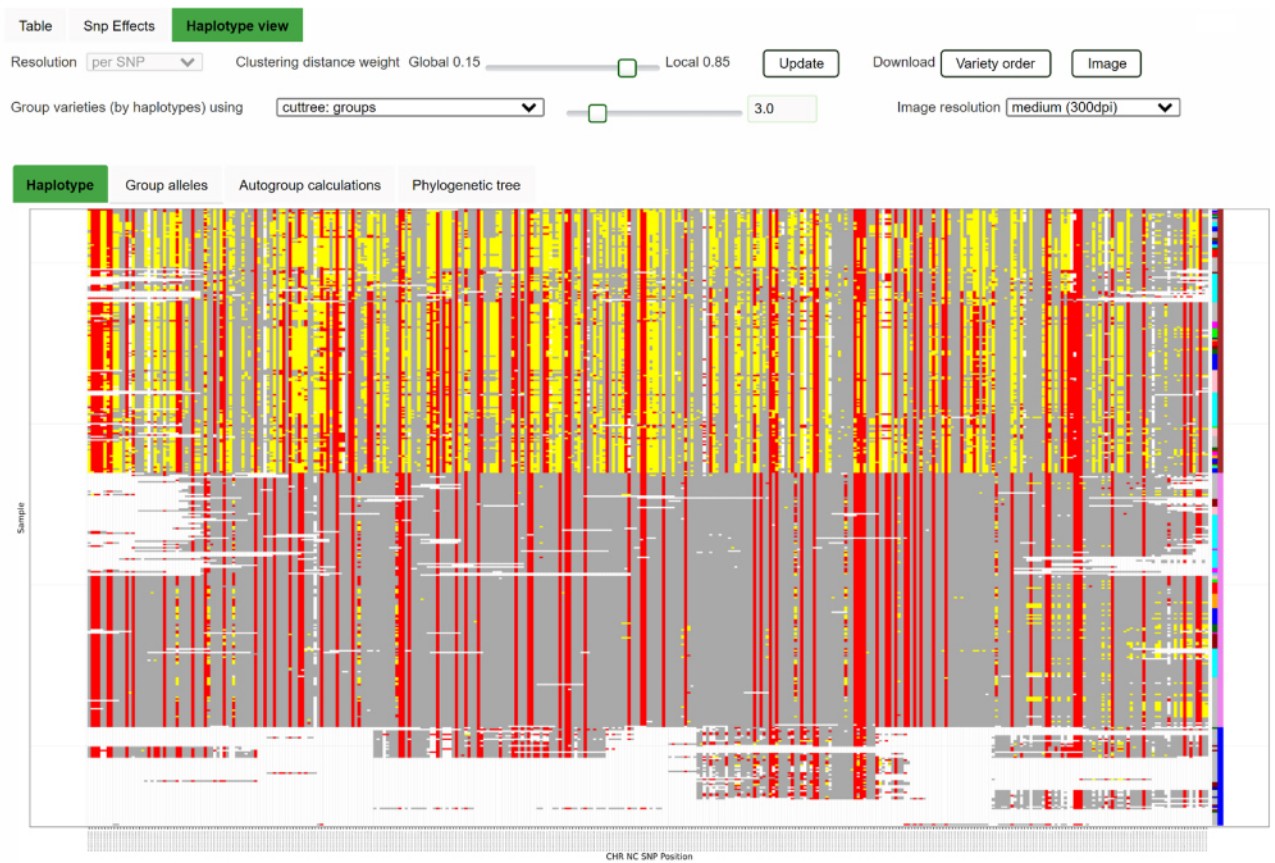

**Figure 3. CannSeek Haplotype viewer.**
The Haplotype viewer of CannSeek clusters the samples of the queried region and displays the heatmap of allele values. In this example, it used the WGS7DS dataset, cs10 reference in the region of LOC115697762 (XP_030480746.1 cannabidiolic acid synthase), chromosome 7 30.98Mb. The heatmap colors indicate reference alleles (gray), alternate (red), heterozygous (yellow), and missing (white). The legacy cultivar type information and cluster group are colored in the last columns. This particular region shows a good match of haplotype cluster and hemp/drug type.

software (RRID:SCR_001905) and the result made available in Newick format. The Python Toytree library was used to visualize the tree and blocks and to export them into SVG format.

With the phylogeny and allele matrix, samples are ordered using the tree leaves order, and the heatmap of alleles is plotted beside the tree. Tree leaf labels are color-coded using available sample types (type I, I, III, or hemp/marijuana). Alleles are color-coded based on their match with the reference (gray: reference homozygous; red: alternate homozygous; yellow heterozygous; white: missing). The scripts are available in the ICGRC Data Generation Protocol step 17 [53], which can be run as a Jupyter Notebook (RRID:SCR_018315).

### SNP Concordance/Discordance

The SNP sites that are unique and common between two datasets are identified using the bcftools stats. Unique and common SNP sites were identified using these dataset pairs: wgs7ds-26trichs, wgs7ds-phylos, and 26trich-phylos. A site discordance analysis was performed using the VCFtools (RRID:SCR_001235) [54] comparison options (gzdiff, diff-site-discordance). Then, the output diff.sites_in_files files were processed to count the



**Figure 4.** **Distribution of SNP attributes for WGS7DS using (a) cs10, (b) pkv5, or (c) fnv2 references.**
SNP properties (left to right): **Call rate**, **MAF**, **Fraction heterozygous**, and **-log(HWE)** distributions of the WGS7DS dataset, including all (**gray**), hemp-type (**blue**), and drug-type (**red**) samples. The step lines are the cumulative count when filtering from the most relaxed (maximum, no filter) to the most conservative (minimum) for quality control. The properties were counted on the vcf file using bcftools +fill-tags, and the histogram was plotted using Python matplotlib.

allele concordance or discordance for each site. Concordance was computed from allele matches, while discordance was computed from allele mismatches for a given SNP site and sample.

Validation of the SNP datasets generated from this study was done by comparing them against publicly available genotype files that use a common set of samples. For datasets having common samples, VCFtools with options *gzdiff, diff-site-discordance*, and *diff-inv-discordance* was used. Then, the concordance/discordance distribution by site and sample were plotted. Two comparisons were made: wgs7ds-woods2022 using the IPK samples on cs10 from Woods *et al.* [33], and wgs7ds-RevGen using PHYLOS samples on pkv5 from Rev Genomics [55].



**Figure 5.** **Distribution of more SNP attributes for WGS7DS using (a) cs10, (b) pkv5, and (c) fnv2 references.**
SNP properties (left to right): **log(Read Depth DP) rate**, **log(Quality)**, and **-log(ExcHet)** distribution and **Genotype vs. Allele Frequency (HWE)** plots of the WGS7DS dataset, including all (**gray**), hemp-type (**blue**), and drug-type (**red**) samples. The step lines are the cumulative count when filtering from the most relaxed (maximum, no filter) to the most conservative (minimum) for quality control. The properties were counted on the vcf file using bcftools +fill-tags, and the histogram was plotted using Python matplotlib.

### *Genome-wide phylogenetic tree*

To generate a whole-genome phylogeny, the SNP datasets were strictly filtered using bcftools with these include parameters (F_MISSING<0.5 & MAF>0.2 & QUAL>100000 & (ExcHet>0.5 | HWE>0.5)), and then LD pruned using bcftools +prune with parameters -m 0.8 -w 100kb. Two tree construction methods were tried, FastTreeMP (RRID:SCR_015501) [56] based on maximum likelihood, and identity-by-state (hamming) distance matrix using plink (distance 1-IBS) followed by neighbor joining. Python Toytree library was then used to lay out the resulting Newick files into circular phylogenetic trees.

**Table 3.** BUSCO scores obtained using the Viridiplantae lineage of predicted PK PKv5 and FN FNv2 genes from FINDER predictions.

| BUSCO output | Purple kush, PKFD | Finola, PKFD |
|---|---|---|
| Gene loci | 41,971 | 33,704 |
| Complete BUSCOs (C) | 268 (63%) | 240 (56.4%) |
| Complete and single-copy BUSCOs (S) | 159 (37.4%) | 208 (48.9%) |
| Complete and duplicated BUSCOs (D) | 109 (25.6%) | 32 (7.5%) |
| Fragmented BUSCOs (F) | 92 (21.6%) | 138 (32.5%) |
| Missing BUSCOs (M) | 65 (15.4%) | 47 (11.1%) |
| Total BUSCO groups searched | 425 | 425 |

### *Multi-dimensional Scaling (MDS) plot*

The WGS7DS, 26TRICH, and Phylos datasets were merged using bcftools, and then filtered to include only SNPs present in both Phylos and WGS7DS. Identity-by-descent distance matrix and MDS coordinates with five components were calculated using TASSEL v5 (RRID:SCR_012837) [57]. The first three components were plotted using a 3D scatter plot of Python plotly library, together with sample information. JavaScripts that allow searching and highlighting data points were then added.

## RESULTS

## Characterization of generated SNPs

The implementation of this section is part of Protocol 1 in [53]. Gene models already available from NCBI Refseq were used for the cs10 genome. For the unannotated PK and FN genomes, FINDER [58] was used to predict gene models, using their genome-specific publicly-available mRNA-Seq sequences as evidence described in the ICGRC Data Generation Protocol step 2 [52]. BUSCO (RRID:SCR_015008) [59] scores of predicted genes using the viridiplantae lineage are shown in Table 3. SnpEff (RRID:SCR_005191) [60] was then run using the gene models to annotate effects of SNPs on gene functions. Specifically, non-synonymous and synonymous SNPs were tagged for visualization and filtering.

In addition to the whole samples set in WGS7DS, separate hemp-type and drug-type sample subsets were generated, and their SNP property distributions were recalculated. Concordance between datasets was determined using bcftools stats. In the CannSeek viewer, the SNP data made available are unfiltered to be flexible to different research objectives of end-users.

## SNP discovery between datasets and references

Table 4 summarizes the number of SNPs and indels discovered for each dataset and reference genome used, and the number of non-synonymous SNPs for the gene models used. The number of SNPs in the WGS7DS set ranges from 91M for cs10 and 109M for fnv2 to 118M for pkv5. The relative number of SNPs for the WGS7DS set against the three references was not as expected. The number of SNPs against fnv2 was between those against cs10 and pkv5 when cs10 and pkv5 are supposed to be more related. The SNP property (Call rate, Minor Allele Frequency, Fraction heterozygous, and HWE p-value) distributions in Figure 4 show how the sample types and references affect the SNP counts. Each histogram plots three groups: all WGS7DS samples, including unknown types (gray), the hemp type in WGS7DS (blue), and the drug type in WGS7DS (red). The discrepancies

**Table 4.** Summary of the raw SNPs and indels discovered.

| WGS7DS | cs10 | Pkv5 | Fnv2 |
|---|---|---|---|
| Samples | 383 | 383 | 381 |
| Records | 104M | 134M | 126M |
| SNPs | 91M | 118M | 109M |
| Indels | 17M | 22M | 21M |
| Multiallelic sites | 24M | 30M | 24M |
| Multiallelic SNP sites | 8M | 10M | 8M |
| Non-synonymous SNPs | 1,334,787 | 956,874 | 523,839 |
| Synonymous SNPs | 1,099,894 | 753,159 | 471,842 |
| **PHYLOS** | **cs10** | **Pkv5** | **Fnv2** |
| Samples | 2216 | 2202 | 2172 |
| Records | 323,953 | 225,422 | 168,956 |
| SNPs | 297,603 | 205,798 | 146,936 |
| Indels | 59,068 | 41,736 | 40,511 |
| Multiallelic sites | 94,527 | 65,399 | 37,374 |
| Multiallelic SNP sites | 26,992 | 18,796 | 17,516 |
| Non-synonymous SNPs | 26,221 | | |
| Synonymous SNPs | 14,778 | | |
| **26TRICH** | **cs10** | **Pkv5** | **Fnv2** |
| Samples | 26 | 26 | 26 |
| Records | 6,274,992 | 397,1367 | 7,424,449 |
| SNPs | 5,201,244 | 3,105,123 | 5,808,480 |
| Indels | 1,107,288 | 355,583 | 508,952 |
| Multiallelic sites | 427,400 | 35,583 | 508,952 |
| Multiallelic SNP sites | 31,288 | 18,614 | 32,700 |
| Non-synonymous SNPs | 361,824 | | |
| Synonymous SNPs | 437,756 | | |

**Table 5.** Concordance between NGS datasets using cs10 reference.

| | WGS7DS vs. 26TRICH | 26TRICH vs. PHYLOS | WGS7DS vs. PHYLOS |
|---|---|---|---|
| Set A | WGS7DS | 26TRICH | WGS7DS |
| Set B | 26TRICH | PHYLOS | PHYLOS |
| Positions unique to set A | 76,052,660 | 5,105,271 | 79,671,208 |
| Positions unique to set B | 1,423,190 | 269,790 | 206,257 |
| Common positions (i+ii+iii+iv+v) | 3,706,099 | 24,018 | 87,551 |
| (i) Biallelic concordant | 3,098,083 | 21,213 | 60,812 |
| (ii) Multiallelic concordant | 16,813 | 34 | 717 |
| (iii) Multiallelic partial concordant | 490,670 | 1,713 | 16,270 |
| (iv) Biallelic discordant | 94,142 | 1,013 | 8,949 |
| (v) Multiallelic discordant | 6,391 | 45 | 803 |
| Discordant/concordant (iii+iv+v)/(i+ii+iii) | 0.164 | 0.121 | 0.334 |

observed were investigated using these plots, as explained in the Discussion section. More SNP property (read depth, quality, and excess heterozygosity p-value) distributions, and Genotype frequency vs. Allele frequency (HWE) are plotted in Figure 5.

For 26TRICH, which are mostly drug samples (23 of 26), 5.2M cs10, 3.1M pkv5, and 5.8M fnv2 SNPs are the expected relative numbers. For the Phylos samples, the SNP numbers are 297k for cs10, 205k for pkv5, and 147k for fnv2, the relative number between references was not expected since these are commercial samples; hence, most are drug-types and should have the most variations against fnv2.

The number of common and unique SNPs between the three datasets (WGS7DS, 26TRICH, PHYLOS) is summarized in Table 5. Using the cs10 reference, the number of shared SNPs



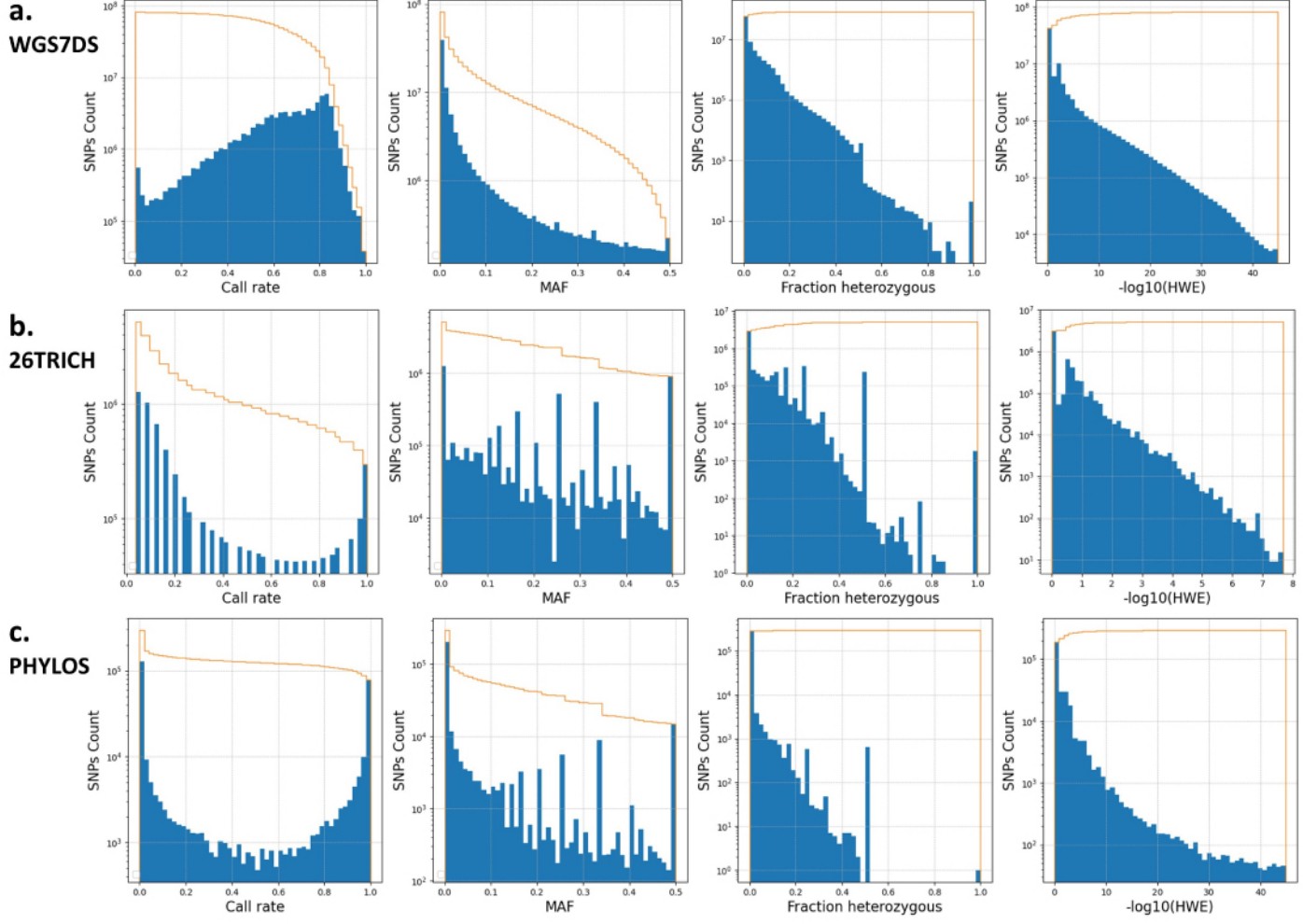

**Figure 6.** **Distribution of SNP attributes for (a) WGS7DS, (b) 26TRICH, and (c) PHYLOS dataset using cs10 reference.**
SNP count distribution (blue) of SNP properties including (left to right) **Call rate**, **MAF**, **Fraction heterozygous**, and **-log(HWE)**. The line (**yellow**) is the cumulative count when filtering from the most relaxed (maximum, no filter) to the most conservative (minimum) for quality control. The properties were counted on the vcf file using bcftools +fill-tags, and the histogram was plotted using Python matplotlib.

between the sample sets was compared. For the WGS7DS, 72% of 26TRICH were in WGS7DS, but only 30% of PHYLOS were in WGS7DS, and 8% of PHYLOS were in 26TRICH. For the common SNPs, the discordant/concordant ratio was 0.16 for WGS7DS vs. 26TRICH, 0.12 for 26TRICH vs. PHYLOS, and 0.33 for WGS7DS vs. PHYLOS. The SNP property distributions for the three datasets using the cs10 reference are compared in the histograms in Figure 6. For the RNA-Seq-derived 26TRICH (b) and amplicon panel PHYLOS (c) datasets, the large number of SNPs with low call rates could be due to ascertainment bias towards restricted regions that may be deleted in many samples. However, when not deleted, these restricted regions have a more uniform minor-allele frequency (MAF) distribution compared to the sharp count decay at high MAFs for the WGS7DS Figure 6a, suggesting that the alleles in the restricted regions are not rare.

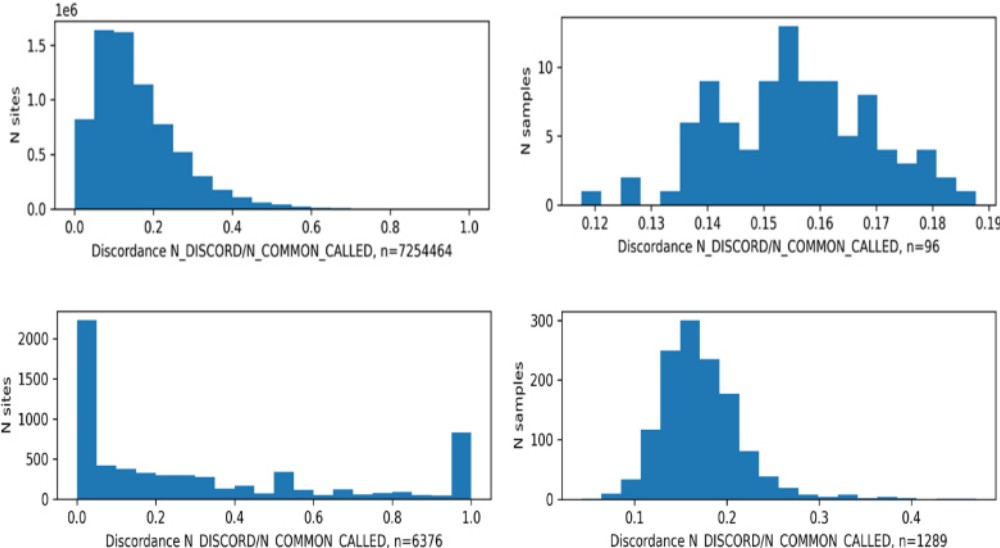

**Figure 7.** **Concordance between (top) IPK samples in WGS7DS vs. Woods *et al.* 2022 using cs10, and (bottom) PHYLOS samples vs. Revgen using PKv5 references.**
With the two vcf files being compared as inputs, discordance was computed using VCFtools ––diff-site-discordance for (left) site discordance, and ––diff-indv-discordance for (right) sample discordance.

**Table 6.** Concordance of SNPs between Cs10 and PK v5 genomes vs. publicly available genotyping results.

| Concordance comparison | Number of SNPs in each comparison |
|---|---|
| **WGS7 vs. WOODS using cs10** | |
| Positions unique to WGS7 | 72,504,295 |
| Positions unique to WOODS | 1,219,985 |
| Common positions | 7,254,464 |
| **PHYLOS vs. REVGEN using PKv5** | |
| Positions unique to PHYLOS | 81,443 |
| Positions unique to REVGEN | 1,782 |
| Common positions | 6,673 |

## Concordance with other public variant sets

The variants discovered were compared to publicly available sets. Table 6 lists the number of common and unique SNPs between the sets using the same set of samples. A total of 85% of SNPs from Woods *et al.* [16] are shared with WGS7DS for the IPK samples using the cs10 reference, and 78% in REVGEN are shared with PHYLOS using the PK reference. For allele concordance, Figure 7 (top) plots the concordance of the WGS7DS dataset and Woods *et al.* using the cs10 reference, and Figure 7 (bottom) plots the concordance of the PHYLOS dataset and the result from Rev Genomics using PK. The average site discordant/concordant ratio was between 0.05 and 0.15, which translates to an 85–95% concordance rate. The 90% site concordance rate is typical, based on studies comparing different variant calling pipelines [61, 62].

## Analyses utilizing the discovered SNPs: use-cases

The following examples demonstrate the utility of the SNP database for diversity analyses at the population level with the construction of phylogenetic trees and MDS plots, highlighting the separation of hemp and medicinal cannabis samples. The SNPs of certain

cannabinoid loci combinations can also group the samples into haplotype blocks that match the types. Lastly, an association analysis was performed using available cannabinoid concentration data on selected samples using the plink regression function.

### Phylogenetic tree of hemp and tetrahydrocannabinolic acid (THCA) dominant cultivars

Phylogenetic trees using the WGS7DS SNPs called against all reference genomes are shown in Figure 8. The filtered sets for tree construction using the criteria described above gave 326,775 SNPs for cs10, 169,519 SNPs for pkv5, and 285,186 SNPs for fnv2. Included were samples of different use-types (left) as hemp/drug based on the published classifications summarized in Table 2, and from different project sources (right). The reads used to generate the reference assemblies were also included. For tree construction, the FastTreeMP method was selected, which uses maximum likelihood rather than distance-based methods.

Using the cs10 reference, the tree in Figure 8 (row 1, left) separates the drug (red) and hemp (blue) types, except for some Jamaican Lions, which are type II but labeled hemp here based on the low THC content from the plot in figure 3 of McKernan *et al.* [14] and the US ferals. The US ferals are a diverse group labeled as unknown type but clustered near the hemp groups. Analyses by Woods *et al.* [16] identify the US ferals as drug type, while Busta *et al.* [63] found them as more hemp than drug. Their position in the middle of the tree between the drug and hemp branches reflects this uncertainty. We also observed that the type II samples clustered with the drug type. No hemp/drug annotations were provided by Woods *et al.* [16] but, for the IPK samples, THCA data from Gloerfelt-Tarp *et al.* [64] were used to classify using the legal threshold of 0.3% for hemp/drug.

To check for batch (data source) bias, the same tree is colored by dataset source in Figure 8 (row 1, right). The scattered location of samples from the same source [13–16] in the tree could indicate that the batch effect did not introduce any bias in the hemp/drug classification. The samples from Kannapedia (green) grouped together either because they are all drug type (a reasonable assumption for commercial samples), or since they have very low read coverage compared to the rest. Identifying batch effect in merged NGS sequences is discussed in detail later with reference to other studies specifically investigating this concern.

### Exploring relative genetic distances between all samples

For another clustering, we merged all datasets WGS7DS, 26TRICH, and PHYLOS and performed multidimensional scaling (MDS). The first three components were plotted in a 3D scatter plot color-coded by data source. Given the unavailability of passport information for most samples and the different genotyping coverage and technologies used, the result was a very coarse representation of cannabis diversity using publicly available data. An interactive MDS plot in Figure 9 to explore all samples is available at [65].

The next two use-cases use the ICGRC-omics API described in the documentation [51] to access SNP data for repeated analysis on different genome regions or to access sample information while using a downloaded genotype matrix.

### Haplotypes for cannabinoid synthases

To demonstrate the utility of allele mining, we queried gene loci for cannabinoid synthases in cs10. The goal was to find the subset of loci that would produce the SNP clustering most



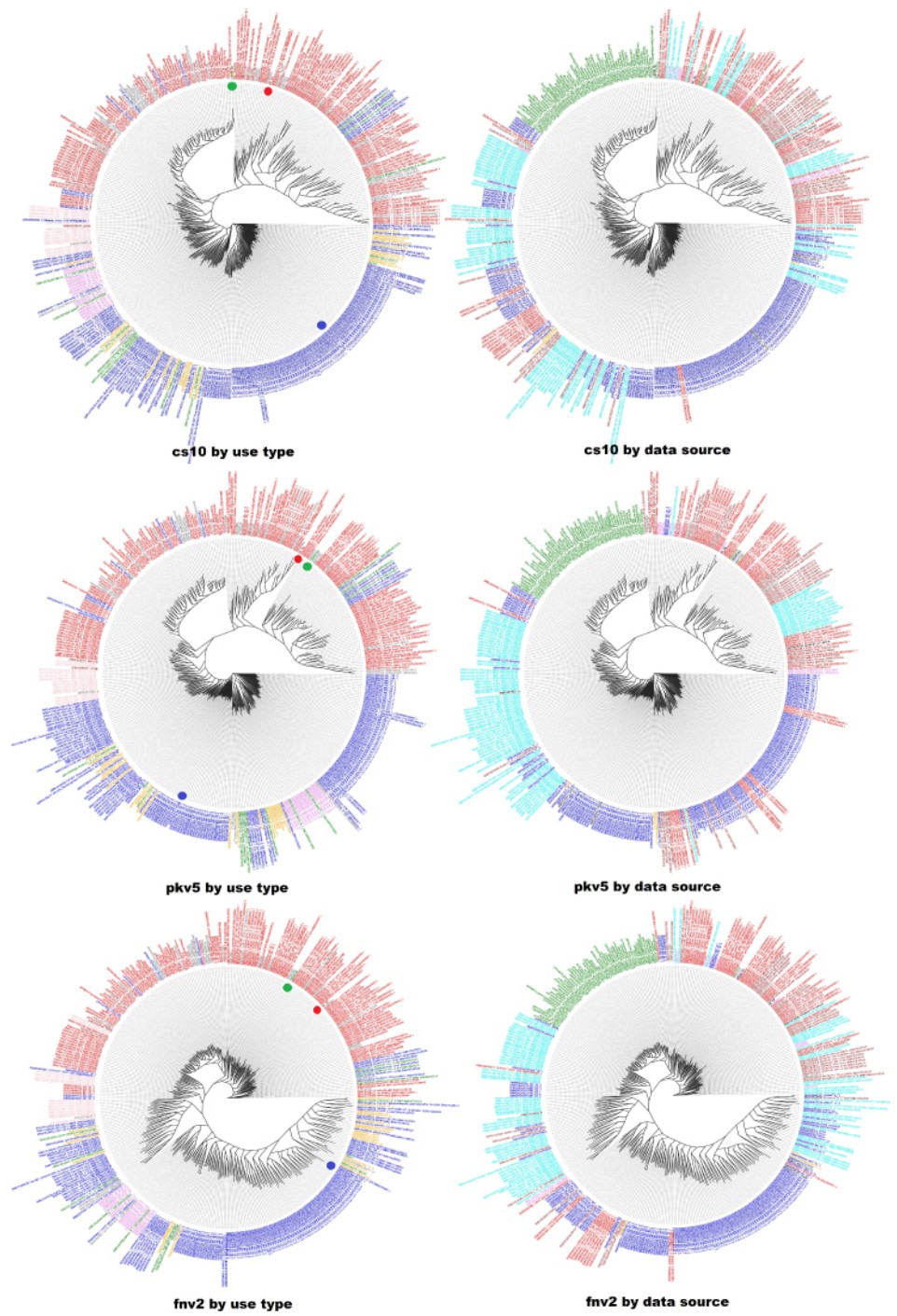

**Figure 8. Phylogenetic tree colored by (left) type and (right) dataset, using three references (row1) cs10, (row2) PK, (row3) FN.**
The trees on the left are colored by use type (red: drug, blue: hemp). Dots indicate the reference genomes (green: cs10, red: PK, blue: FN). On the right, they are colored by data source (brown: McKernan 2020, red: Lynch 2016, blue: Woods 2022, green: Kannapedia, cyan: Ren 2021, orange: assembly projects). The trees are constructed by FastTreeMP using the WGS7DS dataset and the cs10, pkv5 and fnv2 references.

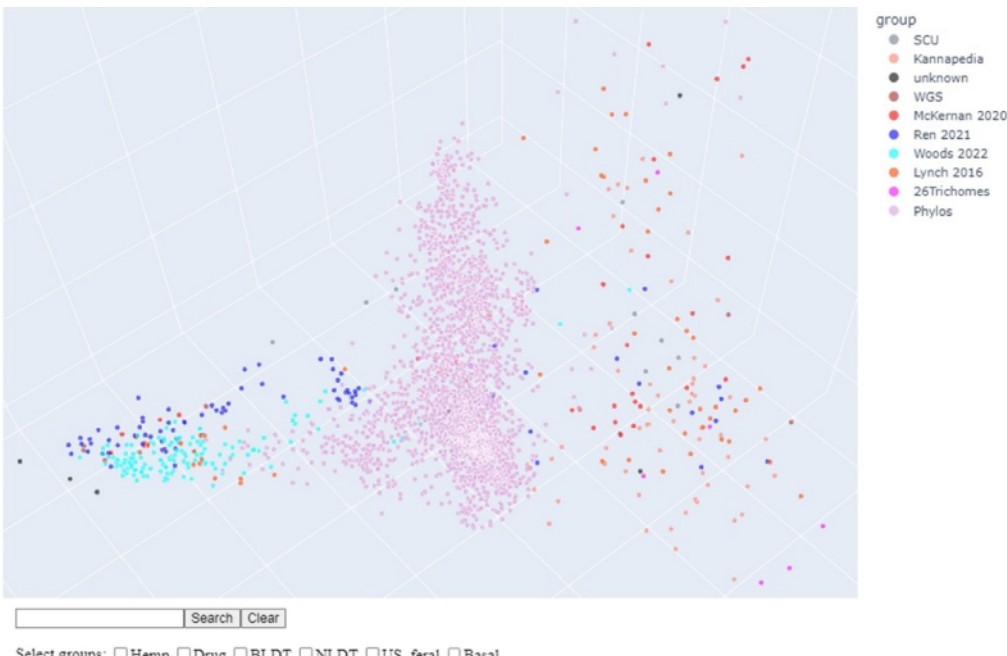

**Figure 9.** **Interactive Multi-dimensional Scaling Plot with all samples.**
Relative genetic distances between all samples using SNPs called against the cs10 reference can be explored using the interactive MDS plot [65]. The samples are colored by data source and can be searched by name or sample type. Hovering over the point shows the sample details. figure-9.html

consistent with the available hemp/drug classification from published sample information. The Jupyter notebook demonstration [66] uses the API to fetch sample information and pre-downloaded filtered genotype data. Using only samples with hemp/drug information, SNPs from all 1-locus, 2-loci, and 3-loci combinations of cannabinoid genes were clustered by Identity by State (IBS) using plink -cluster into two groups ($K = 2$). The distance between the partition between the SNP clustering and the grouping from hemp/drug information was measured using Variation of Information, VI [67]. The different loci subsets were then ranked by VI ––low values indicate better partition matches. Figure 10 shows the (a) best 1-locus, (b) worst, and (c) best loci combination to match the sample types. The best 1-locus clustering was for LOC115697762 (XP_030480746.1, cannabidiolic acid synthase) at 30.9 Mb of chromosome 7 within the region identified in Grassa *et al.* [9]. This is the same locus displayed in Figure 3 using the CannSeek haplotype viewer interface. The best VI was from clustering three gene loci (Figure 10c). The clustering of hemp (blue) and drug (red) samples is evident from the tree in these loci.

### Regression of SNPs against cannabinoid concentrations

Woods *et al.* [16] provide whole-genome resequencing data for the IPK cannabis collection, while Gloerfelt-Tarp *et al.* [64] measured cannabinoid concentrations for samples for the same collection. Although they are neither from the same plant nor sample, and were generated by separate groups, the cultivars came from the same source at IPK. With the assumption that they have identical genotypes, we applied GWAS methods using these data. The number of samples ($N = 84$) was not very large; hence, this did not qualify as a proper

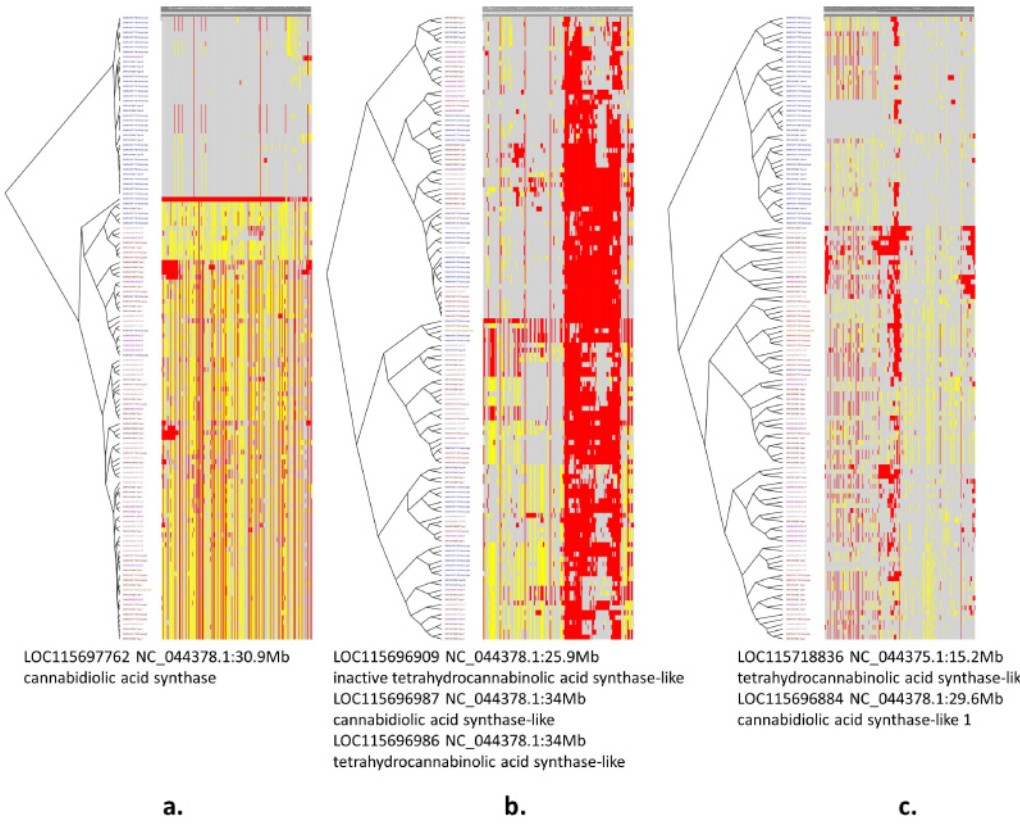

LOC115697762 NC_044378.1:30.9Mb
cannabidiolic acid synthase

LOC115696909 NC_044378.1:25.9Mb
inactive tetrahydrocannabinolic acid synthase-like
LOC115696987 NC_044378.1:34Mb
cannabidiolic acid synthase-like
LOC115696986 NC_044378.1:34Mb
tetrahydrocannabinolic acid synthase-like

LOC115718836 NC_044375.1:15.2Mb
tetrahydrocannabinolic acid synthase-like
LOC115696884 NC_044378.1:29.6Mb
cannabidiolic acid synthase-like 1

a. b. c.

**Figure 10. Cannabinoid synthase loci with (a) best 1-locus, (b) worst, (c) best 3-loci combination partitioning to match the sample hemp/drug annotation.**
Blue/red tree nodes indicate hemp/drug samples. Only samples with hemp/drug information were used. Haplotype clustering was done using different combinations of genomic regions coding for cannabinoid synthases using the omics API and the Jupyter notebook. VI was calculated to measure the match between genotype clusters and sample cultivar type. The heatmap and cluster tree for the best (lowest VI) and worst (highest VI) for 1-, 2-, and 3-loci combinations are shown. The heatmap colors indicate reference alleles (gray), alternate (red), heterozygous (yellow), and missing (white).

GWAS study, but we still identified associations that may guide future investigations (results are viewable at [68]). The Jupyter notebook uses the API to query the phenotypes from Gloerfelt-Tarp *et al.* [64] and the downloadable plink dataset for WGS7DS. The plink-assoc function considers only samples shared in the SNP and phenotype inputs.

## DISCUSSION

While a wealth of DNA resequencing, transcriptome sequencing, and high-density genotyping data, as well as several *de novo* assemblies, are publicly available for *C. sativa*, a comprehensive tool/resource that allows for variant analyses across these datasets relative to key reference genomes is missing. SNP data are widely used for established crop research with wide applications. However, current limitations hinder full utilization, such as the absence of a public gene bank database to host authoritative seed and passport information. Agronomic and omics studies with traceable genotype data are also scarce for robust statistical analysis. Clinical and organoleptic studies are collected by commercial sites, and there are attempts to analyze them [22, 69]. The status is that most phenotypic datasets are kept by commercial entities. An academic or non-government consortium, like

ICGRC, can mediate data sharing and collaborations. The use-cases demonstrated the various applications of these datasets for cannabis. Next, we discuss the uses and concerns regarding this resource.

## WGS7DS SNP properties distribution

The relative numbers of SNPs in the WGS7DS dataset against the three (cs10, fnv2, and pkv5) references were not as expected based on the known reference types. In Figure 4, the Minor Allele Frequencies show SNP spikes near zero, suggesting a high number of rare alleles. Between the hemp and drug types, there is no large difference in the total number of SNPs for all references used at 55M (cs10), 70M (pkv5), and 60M hemp and 70M drug (fnv2). However, the drug type has slightly more (~20%) non-rare (MAF>0.05) SNPs than hemp using the fnv2 reference. With the cs10 and pkv5 references, the number and distribution of SNPs for the hemp and drug types are identical. Excluding the rare alleles SNPs (MAF<0.05), the numbers of SNPs for hemp, drug, and the entire WGS7DS set are identical: approximately 25M against cs10, 35M against pkv5, 30M for hemp, and 35M for drug and WGS7DS against fnv2.

The high number of rare SNPs may be attributed to the unknown sample type, since it includes low-coverage samples and those from miscellaneous sources in small batches. In addition, only ~37M SNPs are shared between the hemp and drug subsets out of the ~54M each type has. With the exponential distribution of MAF, this could mean ~34M SNPs (17M unique to hemp + 17M unique to drug) have potentially low MAF in a combined dataset.

The most conservative threshold for fraction heterozygous (near zero) and -log(HWE) (near zero) already gives a high number of SNPs, confirming the high heterozygosity of the predominantly dioecious cannabis plant [8]. The Call rate distribution can be attributed to the different data sources we used having varying coverage, sample types, and sequencing methods.

In Figure 5, the Read Depth and Quality distributions are identical for the hemp, medicinal, and complete WGS7DS samples. The absence of spikes near zero, or low-quality end, indicates that low-quality calls are minimal with respect to the SNP universe. The counts for low Read Depth for the complete WGS7DS set (gray) are higher than both the hemp (blue) and medicinal types (red) due to the Kannapedia samples having low coverage and unknown types. The same observation on higher counts of bad (near zero) ExcHet in the complete set (gray) than the hemp and medicinal samples may be due to these low coverage samples. In the Genotype Frequency vs. Allele Frequency plot, the heterozygous data points (green dots) are mostly below the expected curve (green pq line) for random mating, indicating most of the samples are not wild but products from inbreeding. Although their pedigrees are undocumented, they are the result of clandestine breeding activities in the past century.

## Exploring the MDS clusters

In the interactive MDS plot at [65], the data points are grouped/colored by the data source. The user can hover over the points to show the details. The points can also be searched by name, use-type (hemp/drug), chemovar (type I, II, and III), or the original grouping assigned by the source. The imposing groups clustering are from Ren *et al.* (blue), Woods *et al.* (cyan), and Phylos (pink). However, the few drug-type samples from Ren *et al.* and Woods *et al.* are separated from the rest of the group, which are mostly hemp-type. The opposite is the case



for the McKernan *et al.* (red) and Lynch *et al.* (orange) samples, which are mostly drug-type, with a few hemp-types separating from their group. The basal types are in the middle between the drug and the hemp clusters. The Phylos samples have no type information, but they tend to cluster towards the drug type, which is reasonable since these are commercial cultivars. The Kannapedia samples (salmon) are all grouped into drug clusters, which was expected since they are also commercial cultivars. The trichome RNA-Seq samples are scattered at wider distances than the rest but still follow the drug/hemp sides. This plot can guide researchers in identifying the type and relationships of cannabis samples deposited in NCBI and other public sites, like finding close relatives with phenotype data.

### Effect of using SNPs called against different reference genomes on phylogenetic groups

The phylogenetic trees in Figure 8 show that cs10 and PK grouped with the drug samples, while FN grouped with hemp types. The branches are longer for the drug samples than the hemp samples when using the cs10 (row 1) or PK (row 2) references. The same observation can be made for the hemp samples when using the FN (row 3) reference. This means the resolution of the samples' genetic distances is higher when the reference is of similar types. This is counter-intuitive since the distances are supposed to be larger when the reference cultivar is different from the samples. However, that is only true when comparing the sample with the reference. When the samples are compared against each other, more differences are expected when the reference used is of the same type. That is, the differences or similarities are amplified when the reference is close to the samples being compared.

A new phylogenetic tree was constructed using the new NCBI standard cannabis assembly Pink pepper (released November 2023), as referenced in Figure 11. This tree shows that Pink pepper is of drug-type, either type I or II.

### Integrating NGS sequences from different sources

This study used NGS sequences generated from different projects, with each project representing a batch of samples. Batch effects must be considered and corrected when comparing abundance values from different batches or sources. In RNA-Seq analyses, the expression levels are normalized across samples. However, for genomic DNA sequences, it is unclear how batches affect the clustering of samples. A study by Lou *et al.* [70] identified that differences in these parameters can cause bias for low-coverage sequencing:

(i)   sequencing chemistry leading to the presence/absence of poly-G tails
(ii)  levels of miscalibration in base quality scores
(iii) read type and read length leading to reference bias/alignment error
(iv)  levels of DNA degradation
(v)   sequencing depths

Quality metrics to identify batch effects aided by Principal Component Analysis (PCA) and filters were developed by Tom *et al.* [71] to remove variants that cause false associations due to batch effects. The standard method to detect a batch bias is to cluster or perform a PCA and color the sample points by batch/data source. Bias is suspected if the clusters coincide with the sample colors. Difficulties arise when the different batches also

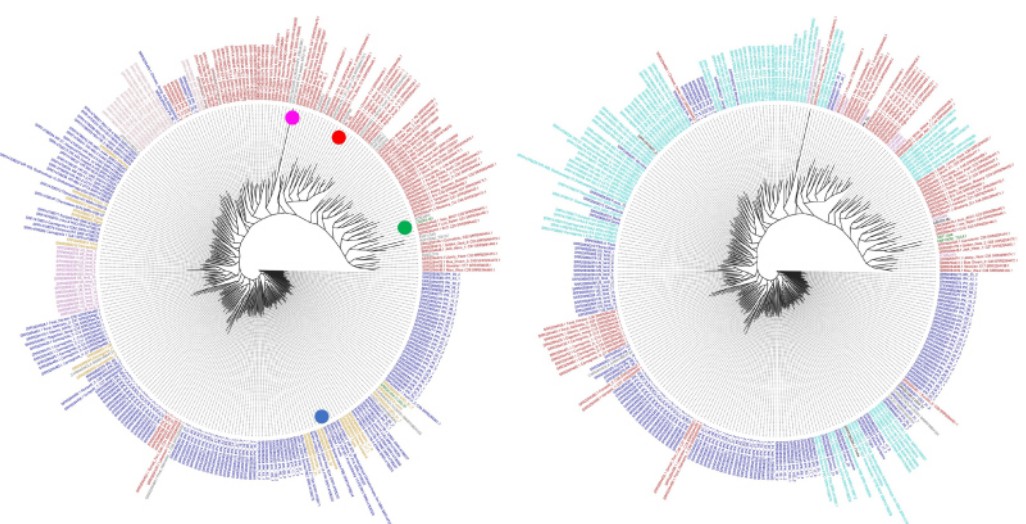

**Figure 11. Phylogenetic tree colored by type (left) and dataset (right) using Pink pepper as reference.**
The tree was constructed using FastTreeMP, as in Figure 8 using WGS7DS plus Pink pepper without the McKernan *et al.* samples, and using Pink pepper as the reference. The trees on the left are colored by use type (red: drug; blue: hemp). Dots indicate the reference genomes (green: cs10; red: PK; blue: FN; pink: Pink pepper). On the right, they are colored by data source (red: Lynch 2016; blue: Woods 2022; green: Kannapedia; cyan: Ren 2021; orange: assembly projects).

belong to different populations, making it challenging to identify if the cluster is due to the batch or the population. This is the situation in some of the data sources used here, with an unbalanced representation of drug or hemp-type samples. Whether the coverage here is high enough to be less affected by the factors mentioned is unclear. Lou *et al.* [70] suggested that bias can still arise for high-coverage data, especially when it comes to accurate calling at low-frequency SNPs. However, differences in sequencing depth are unlikely to be an issue when there is >20× coverage in all batches, but 5–20× may still cause batch effects. The average depths for the data sources used in WGS7DS are plotted in Figure 12. Most are below 10× coverage, suggesting that batch effects may be present.

We tried to identify which data sources were appropriate to combine. We used chi-square statistics to test if genotype clusters are independent from data sources before testing the dependence of a cultivar type on genotype cluster. This requires that all values of the variable should be represented in each data source. This is usually not the case since many studies use either all hemp or all drug-type samples, but not both. We strongly suggest that researchers include at least a few samples of the other cannabis types, even though their main study focuses on one type only.

Despite these limitations, genotype sequences from different sources may be merged with caution, and tests of independence may be performed before merging. The tree successfully separated hemp and drug types; however, there is some doubt due to the non-uniform sample type distribution for most data sources, i.e., some sources consist mostly of one type only. For the batch combination, where the samples show association and the independence hypothesis cannot be rejected (PRJNA310948 and PRJNA575581 with chi-sq *p*-value = 0.3 [13, 14]), the resulting tree shows that hemp and drug types are still separate in the demonstration presented at [72]. All other batch combinations were found to have no sample association (chi-sq *p*-value < 0.05).

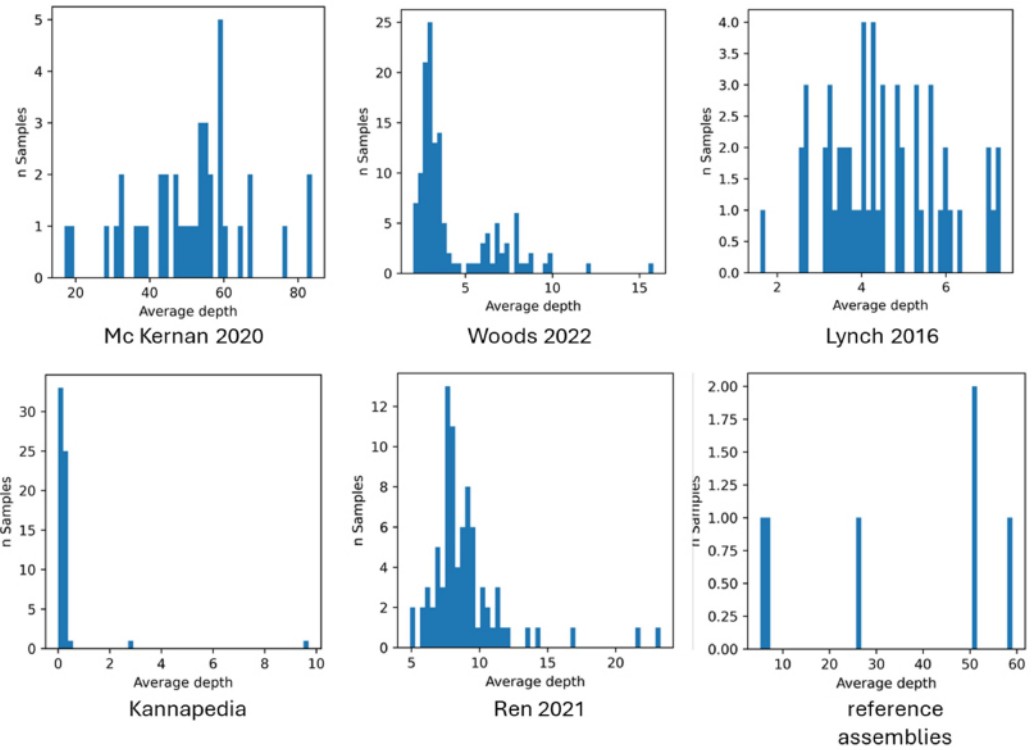

**Figure 12. Average sequencing depth of samples from the different sources used in WGS7DS.**
The values are calculated using bcftools stat -s, and the histograms are plotted separately for each source.

## Incremental update of variants database and established protocols

The GATK GenomicsDB used to merge the GVCF files is a database of variants that supports the incremental addition of samples for joint genotyping. Starting from these databases for several reference genomes and with the pipelines created, the task of classifying new samples and locating their place in the phylogenetic trees can be fast and robust.

The developed pipeline is a result of benchmarking GATK against Parabricks, which we performed as part of the Australia BioCommons bioinformatics working group [73] while generating this database [39]. As a result of benchmarking, we decided to use Parabricks on most analyses due to its speed and service unit savings when using high-performance computing facilities. An issue encountered when using GATK GenomicsDB, related to its change from using a dot (.) to a 0 for representing missing alleles since GATK v4.2.3, was identified and corrected later, as described in this blog [74].

With the wide adoption of NGS technology in cannabis research, we anticipate that more sequencing data will be publicly available. If funding permits, we intend to periodically reanalyze, integrate, and host these results in CannSeek.

## AVAILABILITY OF SOURCE CODE AND REQUIREMENTS

The codes consist of the analysis workflow for SNP discovery and the CannSeek web server derived from Rice SNP-Seek.

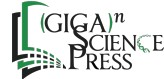

### SNP discovery

- Project name: GATK-Parabricks Gadi Benchmarking
- Project home page: https://github.com/Southern-Cross-Plant-Science/GATK-Parabricks_benchmarking_Gadi_NCI
- Operating system(s): Linux
- Programming language: bash
- Other requirements: PBS HPC job scheduler (for HPC), GATK (CPU) or Parabricks (with GPU)
- License: GNU General Public License Version 2.0
- DOI: 10.5281/zenodo.8348884.

### CannSeek web server (forked from Rice SNP-Seek)

- Project name: CannSeek
- CannSeek project home page: https://github.com/Southern-Cross-Plant-Science/CannSeek
- The startup and loading scripts are available at
  https://github.com/Southern-Cross-Plant-Science/CannSeekBackend.
- Operating system(s): Linux
- Programming language: Java, R, Python
- Other requirements: Apache Tomcat, Postgres (ZK, Spring Framework for development)
- License: Mozilla Public License Version 2.0
- DOI: 10.5524/102571.

### DATA AVAILABILITY

The samples and sources used to generate the SNPs database are listed in Table 1. Their NGS sequences were downloaded from NCBI SRA, or from Kannapedia (https://medicinalgenomics.com/kannapedia-fastq) with the accessions/URLs listed in Samples in [44]. The cs10 CBDRx reference assembly, gene models, and sequences are from NCBI RefSeq GCF_900626175.2. Other cannabis assemblies used from NCBI are PK (pkv5, GCF_900626175.2), FN (fnv2, GCA_003417725.2), and Pink pepper (ppv1, GCF_029168945.1).

pkv5 and fnv2 gene predictions were generated using FINDER, as described in ICGRC protocol [53]. The predicted gene models are in step02-pkfdv1.gff and step02-fnfdv1.gff in [75].

These data files are available in Files in the GigaDB [44]:

- Intermediate files resulting from GigaDB Protocol 1 are ready for loading to the CannSeek database
- CannSeek Tomcat web server Docker/podman image with compile war file, internal paths pointing to host volumes.
- Archived directories to be mapped as Docker/podman volumes for the a. Postgres data, b. Tomcat webapps, c. flatfile directory for HDF5 files, utility scripts, and library.
- The Postgres database populated with the genes and cs10_26TRICH SNP dataset
- HDF5 files for 26TRICH SNPs dataset.

### ABBREVIATIONS

26TRICH, trichome RNA-Seq 26 samples; CBD, cannabidiol; FN, Finola; GWAS, genome-wide association studies; IBS, Identity by State; IPK, German Genebank; MAF, minor-allele

frequency; MDS, multidimensional scaling; NGS, next generation sequencing; PCA, Principal Component Analysis; PK, Purple Kush; SNP, single nucleotide polymorphism; SRA, Sequence Read Archive; THC, tetrahydrocannabinol; THCA, tetrahydrocannabinolic acid; VI, Variation of Information; WGS7DS, whole genome sequencing-7 sources.

## DECLARATIONS

### Ethical approval

The authors declare that ethical approval was not required for this type of research.

### Competing interests

The authors declare that they have no competing interests.

### Author contributions

LM implemented and modified the architecture and contents of the software, and is the primary author of the manuscript. RM supervised the software development and contributed to the development of the manuscript. KLM contributed to the development of the manuscript. TK supervised the overall development of the manuscript as project leader. All authors proofread the manuscript.

### Funding

This study was funded by the Australian Research Council (ARC) Linkage project LP210200606. In addition, first author LM received a stipend from Southern Cross University (SCU). The ICGRC and CannSeek web servers are hosted and funded by SCU.

### Acknowledgements

The authors acknowledge the provision of computing and data resources provided by the Australian BioCommons Leadership Share (ABLeS) program. This program is co-funded by Bioplatforms Australia (enabled by NCRIS), the National Computational Infrastructure and Pawsey Supercomputing Centre.

We thank NVIDIA for the technical support and the provision of the academic license of Parabricks used in this study.

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
