## [Editor Report]

Editor’s AssessmentWith the wide adoption of next generation sequencing (NGS) technology in cannabis research, more and more genomics data is becoming publicly available to improve crop research. This study integrates NGS sequences generated from different projects to provide a comprehensive tool/resource that allows for variant analyses across these datasets. The CannSeek web portal presenting here being built using the Rice SNP-Seek infrastructure with improvements to allow multiple reference genomes and provide a web-service API. The tools built into the portal allow phylogenetic analyses, varietal grouping and identifications, and favorable haplotype discovery for cannabis accessions bringing together curated public sequencing data. Peer review adding additional case-studies to demonstrate the utility of this platform.Editor’s AssessmentWith the wide adoption of next generation sequencing (NGS) technology in cannabis research, more and more genomics data is becoming publicly available to improve crop research. This study integrates NGS sequences generated from different projects to provide a comprehensive tool/resource that allows for variant analyses across these datasets. The CannSeek web portal presenting here being built using the Rice SNP-Seek infrastructure with improvements to allow multiple reference genomes and provide a web-service API. The tools built into the portal allow phylogenetic analyses, varietal grouping and identifications, and favorable haplotype discovery for cannabis accessions bringing together curated public sequencing data. Peer review adding additional case-studies to demonstrate the utility of this platform.

---

## [Reviewer Report]

Reviewer name and names of any other individual's who aided in reviewerZhuang YangDo you understand and agree to our policy of having open and named reviews, and having your review included with the published manuscript. (If no, please inform the editor that you cannot review this manuscript.)YesIs the language of sufficient quality?YesPlease add additional comments on language quality to clarify if neededIs there a clear statement of need explaining what problems the software is designed to solve and who the target audience is? YesAdditional CommentsIs the source code available, and has an appropriate Open Source Initiative license <a href="https://opensource.org/licenses" target="_blank">(https://opensource.org/licenses)</a> been assigned to the code?YesAdditional CommentsAs Open Source Software are there guidelines on how to contribute, report issues or seek support on the code?YesAdditional CommentsIs the code executable?Unable to testAdditional CommentsThis is a database, i can't test its code. But i can work well.Is installation/deployment sufficiently outlined in the paper and documentation, and does it proceed as outlined?Unable to testAdditional CommentsIs the documentation provided clear and user friendly?YesAdditional CommentsIs there enough clear information in the documentation to install, run and test this tool, including information on where to seek help if required?YesAdditional CommentsIs there a clearly-stated list of dependencies, and is the core functionality of the software documented to a satisfactory level?YesAdditional CommentsHave any claims of performance been sufficiently tested and compared to other commonly-used packages? NoAdditional CommentsIs test data available, either included with the submission or openly available via cited third party sources (e.g. accession numbers, data DOIs)?NoAdditional CommentsAre there (ideally real world) examples demonstrating use of the software? NoAdditional CommentsIt will be better to add a case study to show the utility of the database. How to find useful and unique information in the database?Is automated testing used or are there manual steps described so that the functionality of the software can be verified?YesAdditional CommentsAny Additional Overall Comments to the Author1. The download module is not found on the website. I hope the author can provide the download interface. The data displayed in this article are provided in this module, allowing users to download the corresponding data and conduct more personalized analysis according to their own needs. 2. If there is a corresponding relationship between genotype data and phenotypic data, I recommend that the author add a GWAS display to the website. This is very important for breeders looking for phenotypic related SNPs 3. When using GWAS results to find genes, LDblock analysis is a very important reference. It is recommended that the author add an LDblock tool to the website, which allows users to perform LDblock analysis on specific intervals of population genotypes. 4. In the tool module, there is only one blast. Can additional tools be added to make the database more usable? 5. Although there are many data in the database, the data sources, process, and data quality are unknown. And users are not easy to find useful information in the database. The authors should be listed the data sources and publications on help page. This also acknowledges the colleagues of data production.RecommendationMinor Revisions

---

## [Reviewer Report]

Reviewer name and names of any other individual's who aided in reviewerTong Wei, Wenhui ZhouDo you understand and agree to our policy of having open and named reviews, and having your review included with the published manuscript. (If no, please inform the editor that you cannot review this manuscript.)YesIs the language of sufficient quality?YesPlease add additional comments on language quality to clarify if neededIs there a clear statement of need explaining what problems the software is designed to solve and who the target audience is? YesAdditional CommentsIs the source code available, and has an appropriate Open Source Initiative license <a href="https://opensource.org/licenses" target="_blank">(https://opensource.org/licenses)</a> been assigned to the code?NoAdditional CommentsIt is a website without source codesAs Open Source Software are there guidelines on how to contribute, report issues or seek support on the code?YesAdditional CommentsIs the code executable?YesAdditional CommentsIs installation/deployment sufficiently outlined in the paper and documentation, and does it proceed as outlined?YesAdditional CommentsIs the documentation provided clear and user friendly?YesAdditional CommentsIs there enough clear information in the documentation to install, run and test this tool, including information on where to seek help if required?YesAdditional CommentsIs there a clearly-stated list of dependencies, and is the core functionality of the software documented to a satisfactory level?YesAdditional CommentsHave any claims of performance been sufficiently tested and compared to other commonly-used packages? YesAdditional CommentsIs test data available, either included with the submission or openly available via cited third party sources (e.g. accession numbers, data DOIs)?YesAdditional CommentsAre there (ideally real world) examples demonstrating use of the software? YesAdditional CommentsIs automated testing used or are there manual steps described so that the functionality of the software can be verified?YesAdditional Commentsmajor comments: 1. Sample Information: The description of sample information in the paper and the database is limited, relying mainly on descriptions in publications and NCBI BioSample entries. It would be helpful to have more detailed sample information in future studies. 2. Phenotype Information: The phenotypes in the phenome section lack classification and statistics for the types of phenotypes collected in the samples. Additionally, I am wondering whether it is reasonable to include Expression Viewer in this section. 3. Diversity of Data Sources: The paper mentions the use of NGS sequences from different sources. Are there any batch effects in data from different batches? It is recommended to discuss these impacts in more detail and provide some evaluation. 4. Variant calling and Quality Control: Detailed descriptions of each analysis step are required in producing SNP data. For example, how were differences in reference genomes handled during variant calling? What filtering criteria are used to filter low quality SNPs? 5. Results Presentation: In the results presentation section (1) Include statistical data such as the number of SNPs, coverage, etc., to more comprehensively show the database's features. (2) Further analyze the differences in SNP distribution among different sample types (e.g., industrial hemp, medicinal cannabis) to reveal their genetic variation characteristics. (3) It would be nice to see some examples, such as SNP-based genetic diversity analysis and association analysis, to showcase the database's applicationAny Additional Overall Comments to the AuthorThe creation of the CannSeek is a significant contribution to the field of cannabis genomics, filling a gap in existing research tools. It provides a user-friendly interface that allows researchers to access and analyze large amounts of genotype data, which is crucial for genetic research and cultivar improvement of cannabis. The paper details the process of generating, storing, and retrieving SNP data, as well as the related bioinformatics tools and methods, which are very useful for other researchers to replicate and extend these methods. Although some details should be provided to increase the data accessibility.RecommendationMinor Revisions